# Antenatal Glucocorticoid Administration Promotes Cardiac Structure and Energy Metabolism Maturation in Preterm Fetuses

**DOI:** 10.3390/ijms231710186

**Published:** 2022-09-05

**Authors:** Kenzo Sakurai, Yuko Takeba, Yosuke Osada, Masanori Mizuno, Yoshimitsu Tsuzuki, Kentaro Aso, Keisuke Kida, Yuki Ohta, Masanori Ootaki, Taroh Iiri, Isamu Hokuto, Naoki Shimizu, Naoki Matsumoto

**Affiliations:** 1Department of Pediatrics, St. Marianna University School of Medicine, Kawasaki 216-8511, Kanagawa, Japan; 2Department of Pharmacology, St. Marianna University School of Medicine, Kawasaki 216-8511, Kanagawa, Japan

**Keywords:** antenatal glucocorticoid, cardiac growth, fetal heart

## Abstract

Although the rate of preterm birth has increased in recent decades, a number of preterm infants have escaped death due to improvements in perinatal and neonatal care. Antenatal glucocorticoid (GC) therapy has significantly contributed to progression in lung maturation; however, its potential effects on other organs remain controversial. Furthermore, the effects of antenatal GC therapy on the fetal heart show both pros and cons. Translational research in animal models indicates that constant fetal exposure to antenatal GC administration is sufficient for lung maturation. We have established a premature fetal rat model to investigate immature cardiopulmonary functions in the lungs and heart, including the effects of antenatal GC administration. In this review, we explain the mechanisms of antenatal GC actions on the heart in the fetus compared to those in the neonate. Antenatal GCs may contribute to premature heart maturation by accelerating cardiomyocyte proliferation, angiogenesis, energy production, and sarcoplasmic reticulum function. Additionally, this review specifically focuses on fetal heart growth with antenatal GC administration in experimental animal models. Moreover, knowledge regarding antenatal GC administration in experimental animal models can be coupled with that from developmental biology, with the potential for the generation of functional cells and tissues that could be used for regenerative medical purposes in the future.

## 1. Introduction

Based on aggregated vital statistics data from 2007 to 2017 obtained from the Ministry of Health, Labor, and Welfare in Japan, the rates of low birth weight and preterm infants have increased from 7.8% to 9.4% and from 5.0% to 5.7%, respectively. Although the birth rate is low among advanced countries, premature infants (defined as delivery at <32 weeks of gestational age) are more likely to suffer from respiratory distress syndrome, sepsis, intraventricular hemorrhage, and necrotizing enterocolitis shortly after birth [1,2]. Furthermore, several studies have reported that an increase in the frequency of chronic diseases such as diabetes, obesity, cardiovascular diseases, and hypertensive disorders in adulthood can be attributed to an influence of the intrauterine environment and preterm birth [3,4,5,6,7]. Additionally, preterm birth is associated with elevated cardiovascular morbidity and mortality in adulthood [5,6].

The hypothesis of the Developmental Origins of Health and Disease (DOHaD) was initially formulated by Sarah McMullen and colleagues [8]. Their studies demonstrated an inverse relationship between birth weight and the development of high systolic blood pressure and mortality from ischemic heart disease, with the intrauterine environment as an indirect factor [8]. Subsequently, substantial evidence supporting the DOHaD hypothesis has been obtained from animal models and human studies [8,9,10,11]. For example, traditional cardiovascular-disease risk factors are significantly associated with an increased risk of preterm birth [12,13,14]. However, the exact mechanisms that permanently change the development, structure, and function of vital organs with fetal growth are still poorly understood. The development of the heart is especially a highly dynamic process that can be conceptually divided into morphogenesis, energy metabolism, and functional maturation [14,15].

Many hormones produced by the placenta are essential for proper fetal growth [16]. The hypothalamic–pituitary–adrenal axis plays an important role during embryonic development, with potent programming effects on organ development. An in vitro study showed that steroid hormones, including glucocorticoids (GCs), are essential for cardiomyocyte maturation, accelerating the differentiation from pluripotent stem cells to cardiomyocytes [17].

Endogenous cortisol concentrations (<17.5 ng/mL^−1^) are maintained five- to ten-fold lower than maternal levels during gestation, with the concentrations rising markedly during the last 5–10 days of gestation, in fetal sheep [18,19]. Because preterm births occur before this physiological rise in endogenous GCs, the organs and tissues, which, as a result, were not subjected to GC action, remain immature at birth [18,19].

Antenatal GC treatment is the standard of care for women at risk of preterm labor between 24 and 34 weeks of gestation [20]. GC treatment reduces the rates of neonatal morbidity and mortality and decreases the risk of respiratory distress syndrome and intraventricular hemorrhage after birth [21]. Liggins and Howie published the first randomized controlled trial in humans in 1972, and many relevant studies have followed [22]. Roberts et al. conducted a systematic review of 30 clinical trials; the authors concluded that antenatal corticosteroid treatment was associated with a reduction in the most serious adverse outcomes related to the prematurity of organs requiring mechanical ventilation and systemic infections in the first 48 h of life [21].

Although antenatal GC administration has been verified for lung development, its potential effects on other organs are debatable. Several reports have shown that antenatal GC treatment is beneficial for cardiovascular development [6,23,24,25,26]. Furthermore, preterm fetal exposure to maternal GCs accelerates cardiovascular maturation, improves basal cardiovascular function, and enhances fetal cardiovascular defense against physiological stress in extrauterine life [27]. Conversely, some studies have reported that antenatal GC exposure in young children is associated with changes in glucose metabolism and localized changes in aortic function, increasing the risk of cardiovascular failure when adulthood is reached [28]. Additionally, Ivy et al. reported that antenatal corticosteroid administration may interfere with fetal heart maturation by downregulating the ability to respond to GCs. Furthermore, GCs regulate mitochondrial fatty acid oxidation in fetal cardiomyocytes [29]. Therefore, further research on animal models and human infants is required to clarify the benefits and adverse effects of antenatal GC administration on cardiac function.

Animal models using mice, rats, and sheep have been useful resources in studying the effects of antenatal GC exposure [30,31]. For example, a sheep study showed that fetal lung maturation was induced when pregnant ewes received two intramuscular doses of GC (0.25 mg/kg/dose) 24 h apart [32]. Most animal studies have attempted to imitate antenatal GC administration during late gestation in clinical practice. Studies by Samtani and colleagues described the optimal regimen of dexamethasone (DEX) for fetal lung maturation during the late gestation period, using pharmacokinetic/pharmacodynamics simulation, and reported the effects of a total dose of 6 μM/kg in pregnant rats [33,34]. We established an animal model based on that in several studies, including the studies by Samtani et al. [33,34]; we administered DEX to pregnant rats for 2 days, starting at day 17 of gestation. We found that antenatal GC administration increased pulmonary surfactant in our model rats, indicating lung maturation [35].

In this review, we especially focus on the molecular mechanisms involved in the morphological and physiological growth of the fetal heart in a preterm infant model. We provide ongoing data on the signaling pathway of GC action in the cytoplasm, mitochondria, and sarcoplasmic reticulum (SR) in the heart.

## 2. Cardiac Physiological Hypertrophy by Cardiomyocyte Proliferation with Antenatal GC Administration

In mammals, the proliferation capacity of embryonic and fetal cardiomyocytes predicates the formation of mature heart chambers, and the cardiomyocytes are terminally differentiated. However, these cells rarely divide during adulthood [36]. Preterm myocardial tissue shows a marked reduction in cardiomyocyte proliferation compared to that in full-term infants, providing evidence of a delay in normal cardiomyocyte hyperplastic growth [5]. There is concern that antenatal GC exposure in premature infants may inhibit cardiomyocyte replication and result in a heart with fewer cells than that in full-term infants [36]. Therefore, we have investigated the structural maturity of the myocardium and the signaling pathway involved in growth in fetal rats administered antenatal GC [37,38,39]. In terms of morphology, we have found that antenatal GC administration increases the cross-sectional area of the ventricular myocardium. Although irregular myofibril orientation was observed morphologically in 19-day fetal hearts, myofibril components were organized at 19 days after DEX administration; however, the cause of hypertrophy was uncertain [37,38]. Furthermore, Ki-67-positive cells, a proliferation marker, were significantly increased in the myocardium after DEX administration. These results indicate that cardiac enlargement results from cardiomyocyte proliferation [37,38].

Signaling pathways connecting growth factors and their downstream mediators play a role in the development of physiological cardiac hypertrophy. The phosphatidylinositol-3 kinase (PI3K)–Akt pathway participates in the cardiac hypertrophic program based on heart growth [40,41]; our previous study describes the signaling pathway involved in cardiomyocyte proliferation in greater detail. Antenatal DEX administration has been shown to activate PI3K/phosphorylated glycogen synthase kinase-3β (p-GSK-3β)/β-catenin, which contributes to cardiac growth [38]. Specifically, as shown in Figure 1, GCs activate PI3K, and, subsequently, Akt-1 accelerates both p-GSK-3β and vascular endothelial growth factor (VEGF) expression [38]. Moreover, p-GSK-3β expression induced by Akt protein directly regulates downstream transcriptional factors, such as β-catenin and the zinc finger transcription factor, GATA-4 [42,43]. GATA-4 is related to cardiac myocyte hypertrophy [44,45]. An increase in GATA-4 induces an increase in myocardium contraction-related protein troponin T [37]. In addition, Akt-1-deficient mice show a decrease in endothelial cell density in the myocardium, attenuated expression of VEGF in the fetal heart, and reduced survival. Furthermore, p-GSK-3β protein, which is an important regulator of downstream substrates that are adjusted by Akt, is significantly increased with fetal DEX administration. β-Catenin and VEGF protein levels are also significantly increased. β-Catenin activation robustly enhances the proliferative capacity of early cardiomyocytes. VEGF is involved in regulating coronary angiogenesis during physiological cardiac hypertrophy [45].

In the developing heart, cardiomyocyte proliferation is regulated by several different pathways: the neuregulin/ErbB/ERK pathway is important for primary proliferation in the embryonic heart, the PI3K/Akt/GSK3β pathways are required for hyperplastic growth of cardiomyocytes, and the Hippo-Yes-associated protein (Yap) pathway is a critical regulator of cardiomyocyte proliferation [46,47]. The Hippo pathway is involved in the cardiomyocyte proliferation underlying fetal heart growth. After birth, heart growth switches to an increase in cell size, which is called physiological hypertrophy [48,49]. Inactivation of the Hippo pathway leads to Yap transcription activation, with proliferative gene stimulation by the intranuclear transition of Yap [50]. The Yap protein is clearly detected in neonatal hearts and declines with age, while Yap phosphorylation increases with age [51]. Yap has been shown to regulate cardiomyocyte fate through multiple transcriptional mechanisms. For example, Yap is known to activate Akt in cardiomyocytes [52]. Yap expression in the nuclei of fetal cardiomyocytes under antenatal DEX administration is strongly correlated with cyclin D1- and Ki-67-positive cardiomyocytes. Interestingly, Yap is markedly expressed in the epicardium, which may contribute to not only cell proliferation but also coronary vascular differentiation in the fetus [39,48].

Furthermore, whether GC action is mediated by GC receptors (GRs) or other receptors is important. In an in vitro study, increased cardiomyocyte proliferation after culturing with DEX was shown to be mediated by GRs. Specifically, pre-incubation with the GR antagonist, RU486, significantly inhibited the increase in DEX-induced cardiomyocyte proliferation in 1-day neonatal cardiomyocytes [38]. GRs have isoforms, and the GR β isoform is closely related to increased Akt phosphorylation [53]. On the other hand, Reini et al. reported that mineralocorticoid receptors (MRs), as well as GRs, are involved in fetal heart enlargement due to relevant increases in maternal cortisol levels [48,54]. Specifically, fetal heart enlargement in response to a modest and chronic rise in maternal cortisol levels is mediated by MRs and, to a lesser extent, GRs within the heart [48,54]. These results indicate that the sequence of responses is critically important for mediation through GRs within the heart [48,54].

To summarize the above findings, antenatal DEX induces structural maturity, with accompanying cardiomyocyte proliferation in the premature fetal rat heart, and the Akt/p-GSK-3β pathway and Yap are thought to contribute to cardiac growth with angiogenesis.

## 3. Adenosine Triphosphate (ATP) Production in Glycolysis and Mitochondria Function

Energy metabolism in the fetal heart is regulated in a manner that differs from that in adult cells. The neonatal heart muscle depends on fatty acid metabolism to provide energy for contraction, which is more efficient than glycolysis. The transition from glycolysis to fatty acid metabolism involves a complex process including the maturation of mitochondria and dramatic changes in circulating levels of fatty acids and lactate [55]. Furthermore, fetal cardiomyocytes generally have immature mitochondria, and the fetal heart is relatively more dependent on anaerobic glycolysis [55,56]. In the fetal heart, glucose and lactate are primarily the preferred sources for ATP production. Changes in intermediary metabolism occur during the development of the heart. The enzyme α-enolase is a high-energy intermediate in ATP production that catalyzes the conversion of 2-phosphoglycerate to phosphoenolpyruvate and plays a critical role in the glycolytic pathway of energy production in the fetal heart. Enolase has three subunits, α-, β-, and γ-enolase; the dimer of the α-subtype is widely distributed in embryonic organ tissues, and α-enolase converts glucose into two three-carbon molecules, called pyruvate, which is the end-product of glycolysis. Energy released during glycolysis is used to make ATP [57,58]. The α-enolase levels in the heart are higher in 19-day and 21-day fetuses than in 1-day neonates [59]. Antenatal GC treatment increases both α-enolase production and pyruvate levels; accordingly, ATP and cAMP levels are also significantly increased in 19-day fetuses. These results suggest that α-enolase, which plays a critical role in the glycolytic pathway, is an important factor in fetal circulation [59]. The acceleration in glycolysis may be related to cardiac contraction ability in the fetal myocardium.

Increased cardiac energy metabolism is essential for normal cardiac contractile function. ATP is mainly synthesized in the mitochondria of cardiomyocytes, and creatine kinase (CK) reaction is the prime energy reserve of the heart, providing a rapid source of ATP and enhancing its delivery from mitochondrial sites of production to sites of use, including the myofibrils [60,61,62].

There are multiple isoforms of CK among the three cytosolic isoforms (MM, MB, and BB). Myofibrillar-bound M isoenzyme (CK-M) is predominant in the heart [59]. The downregulation of CK levels has been shown to lead to heart failure in animal models and in humans. Furthermore, a lack of both Mi-CK and CK-M isoforms has been implicated in cardiac dysfunction in mice [63].

After birth, mitochondrial oxidative phosphorylation via Mi-CK is an important process in ATP production [27]. If the functional capacity of Mi-CK is insufficient to produce energy, preterm infants cannot adapt to extrauterine life because of cardiac failure. Antenatal GC administration has been shown to significantly increase the mRNA and protein expression of CK isoforms, CK-M and Mi-CK, in fetal and neonatal rat hearts and accelerate mitochondrial respiration-dependent ATP synthesis in fetal and neonatal hearts [64,65,66]. In addition, peroxisome proliferator-activated receptor (PPAR) γ, upstream of CK gene upregulation, has been shown to contribute to energy metabolism and mitochondrial biogenesis in fetal and neonatal hearts and cultured cardiomyocytes. Antenatal GC administration activates glycolysis and, consequently, mitochondrial function, accelerating ATP synthesis capacity in the immature heart.

## 4. SR Calcium Transport ATPase 2a (SERCA2a) and Phospholamban in the Premature Fetal Rat Heart with Antenatal GC Administration

During fetal and postnatal development, cardiomyocytes become terminally differentiated muscular cells that are connected end-to-end by gap junctions, allowing concerted contractile activity. The cardiac SR plays an indispensable role in the contraction–relaxation sequence. The SR is related to the contraction process of the heart, as Ca^2+^ is taken up and released from the SR [67]. SERCA2a, an energy-requiring calcium uptake pump located in the cardiac SR membrane, contributes significantly to Ca^2+^ homeostasis [68]. Phospholamban is a critical regulator of SERCA2a Ca^2+^ affinity. In the rabbit heart, SR function dramatically changes during the perinatal period, suggesting that adult hearts are most likely dependent on Ca^2+^ release and uptake from the SR, while perinatal hearts are more dependent on trans-sarcolemmal Ca^2+^ influx than on SR Ca^2+^ release in excitation-contraction coupling [69]. SERCA2a begins to contribute to Ca^2+^ homeostasis at an early stage, and the Ca^2+^ content of the SR increases with age. SERCA2a immaturity and the subsequent alteration in calcium cycling results in heart failure [70,71,72].

Antenatal GC administration increases the expression of SERCA2a and phospholamban protein in the fetal heart, resulting in the acceleration of Ca^2+^ cycling in the SR [73]. These processes may lead to the maturation of fetal cardiac function. A schematic diagram of energy production and cardiac contraction acceleration is shown in Figure 2. SERCA2a mRNA levels are lower in 19-day-old and 21-day-old fetal rat hearts than in 1-day-old neonates and adult rats. Additionally, SERCA2a protein levels tend to be lower in 19-day-old and 21-day-old fetal hearts than in neonatal and adult rats [73]. Antenatal DEX administration significantly increases SERCA2a production in the left ventricle (LV) and right ventricle (RV) of 19-day-old and 21-day-old fetal hearts [73]. Additionally, phospholamban protein levels in the heart are significantly increased in 19-day-old and 21-day-old fetuses treated with DEX. Phospholamban accumulation is observed in 1-day-old neonates. Antenatal DEX administration also increases phospholamban production in the RV and LV of 19-day-old and 21-day-old fetal hearts [73]. In contrast, antenatal GC administration does not affect the expression of type 2 ryanodine receptor, a Ca^2+^ transporter from the SR to the intercellular space [71,73]. The resulting increase in intracellular Ca^2+^ concentration has also been verified in cardiomyocytes treated with GC; the GC effect was mediated through cardiomyocyte GRs.

To summarize the above findings, antenatal GC administration increases SERCA2a and phospholamban protein expression levels in the fetal heart, resulting in an acceleration of Ca^2+^ cycling in the SR [73]. These processes may lead to the maturation of fetal cardiac function. Particularly, antenatal GC therapy may contribute to Ca^2+^-handling-mediated development of SR function in preterm infants.

## 5. Cardiac Effects in Clinical Antenatal GC Therapy—From Bedside to Bench

In a review of 18 trials, with data on over 3700 babies, antenatal administration of 24 mg betamethasone, 24 mg DEX, or two grams of hydrocortisone in women at risk for preterm birth was associated with a significant reduction in mortality, respiratory distress syndrome, and intraventricular hemorrhage in preterm infants [74]. These benefits extended to a broad range of gestational ages and were not limited by sex or race. In addition, no adverse consequences of prophylactic corticosteroid use for preterm births have been identified.

Furthermore, antenatal steroid infusion regimens previously shown to have significant pulmonary effects have also been shown to significantly improve cardiovascular adaptation at birth in premature newborn sheep [23]. Cardiovascular effects in animals treated with hydrocortisone for 60 h before delivery include higher blood pressure and cardiac output and improved LV contractility [75]. Additionally, the infusion of both corticosteroids and thyrotropin-releasing hormone improves postnatal blood pressure compared to that in animals treated with corticosteroids alone or control animals [75]. Padbury and colleagues evaluated the effect of a direct fetal injection of betamethasone in preterm sheep 48 h before cesarean delivery and found that, after delivery, steroid-treated animals had significantly elevated postnatal blood pressure, cardiac output, and cardiac contractility, despite similar preload, mean arterial pressures, and calculated systemic vascular resistances, compared to those in control animals [76]. These results suggest a significant direct effect of betamethasone on myocardial contractility [76]. The mechanisms underlying this observation partly involve a significant increase in β-adrenergic-receptor-dependent myocardial cyclic adenosine monophosphate generation [77]. Furthermore, in a study designed to assess the risk of patent ductus arteriosus (PDA), the incidence of symptomatic PDA was significantly reduced in preterm infants of mothers antenatally treated with corticosteroids [78].

Although there are several case reports of prolonged QT interval in adults with adrenal insufficiency, to our knowledge, there are no reported fetal cases [79,80,81]. Previous reports have described cases of Torsade de Pointes associated with hypopituitarism, which was treated with steroid hormones. As premature infants have reduced adrenal function, it could be speculated that antenatal GC administration may affect the QT interval [79,80,81].

Regarding the clinical indications for antenatal GC therapy, it is recommended in pregnant women between 24 0/7 weeks and 33 6/7 weeks of gestation who are at risk for preterm delivery within 7 days, including those with ruptured membranes and multiple gestations [82]. The GC therapy regimen (e.g., the quantity and timing of GC dosing, choice of medicine, etc.) should be adjusted according to composite risk factors in pregnant women. In clinical practice, the indication for betamethasone is currently being expanded; beginning at 23 0/7 weeks of gestation, betamethasone may be considered for pregnant women who are at risk of preterm delivery within 7 days, based on the family’s decision regarding resuscitation. In addition, there are an increasing number of reports on repeated corticosteroid administration. One repeat dose of corticosteroids should be considered for pregnant women who are at less than 34 0/7 weeks of gestation, at risk of preterm delivery within 7 days, and received previous corticosteroid treatment at more than 14 days prior [82].

Animal models are essential for providing convincing evidence regarding a causal relationship between fetal development in early life and increased risk of adult disease in later life [83]. This evidence is enhanced by animal models, narrowing the knowledge gap between animal models and future clinical translation. Pregnancy in women at risk of adult obesity and type 2 diabetes has been replicated in an animal model [84]. However, it is difficult to develop an animal model imitating pregnant women with composite risk factors. Thus, further investigation is needed using animal models of pregnancy with composite factors. Thereby, indications for antenatal GC therapy will be gradually expanded clinically.

Given that basic research suggests antenatal GC administration may contribute to the maturation of premature hearts by accelerating cardiomyocyte proliferation, angiogenesis, energy production, and SR function, antenatal GC therapy may be effective not only in preterm infants but also in fetuses with predicted reduced cardiac function, for improved future prospects.

## 6. Limitations

Animal models of premature infants have provided valuable information on the structural and functional features of the heart [14,15,23,24,69]. Indeed, these models have shown that antenatal GC administration accelerates premature fetal cardiomyocyte proliferation, angiogenesis, energy production, and SR function. Although these models cannot reflect all clinical features, they are valuable for understanding the mechanisms of maturation. However, many issues, including the effects of antenatal GC administration on other organs, other molecular pathways, etc., in preterm fetuses must be addressed while using novel approaches. Future studies need to use establish clinical approaches, such as cardiac functional evaluation, using echocardiography, and myocardial tissue characterization, using cardiovascular magnetic resonance imaging, to support antenatal GC administration in human neonates and animal models.

## 7. Conclusions

Antenatal GC administration has proven to be useful in the clinical field for over 40 years. To provide high-quality perinatal care, evidence regarding the benefits and adverse effects of GC administration in preterm infants must be accumulated in terms of the tissues and organs of the whole body in human and animal model studies. Based on our studies and others, the hallmarks of heart maturation, including the cardiomyocytes, following antenatal GC administration can be discerned. Antenatal GC administration accelerates cardiomyocyte proliferation, angiogenesis, energy production, and SR function, as established in our preterm model rats. Overall, these results indicate that antenatal GC treatment may contribute to the maturation of the premature heart.

Molecular determinants of heart development have received much attention over the past several decades. In this review, we summarized novel information on structural changes, acceleration in energy production, and contraction ability with antenatal GCs in terms of preterm heart maturation. The functions of fetal heart maturation and regulators in this process may lose their ability to function in adulthood. The knowledge regarding antenatal GC administration in experimental animal models can be coupled with that from developmental biology, with potential for the generation of functional cells and tissues that could be used for regenerative medical purposes in the future.

## Figures and Tables

**Figure 1 ijms-23-10186-f001:**
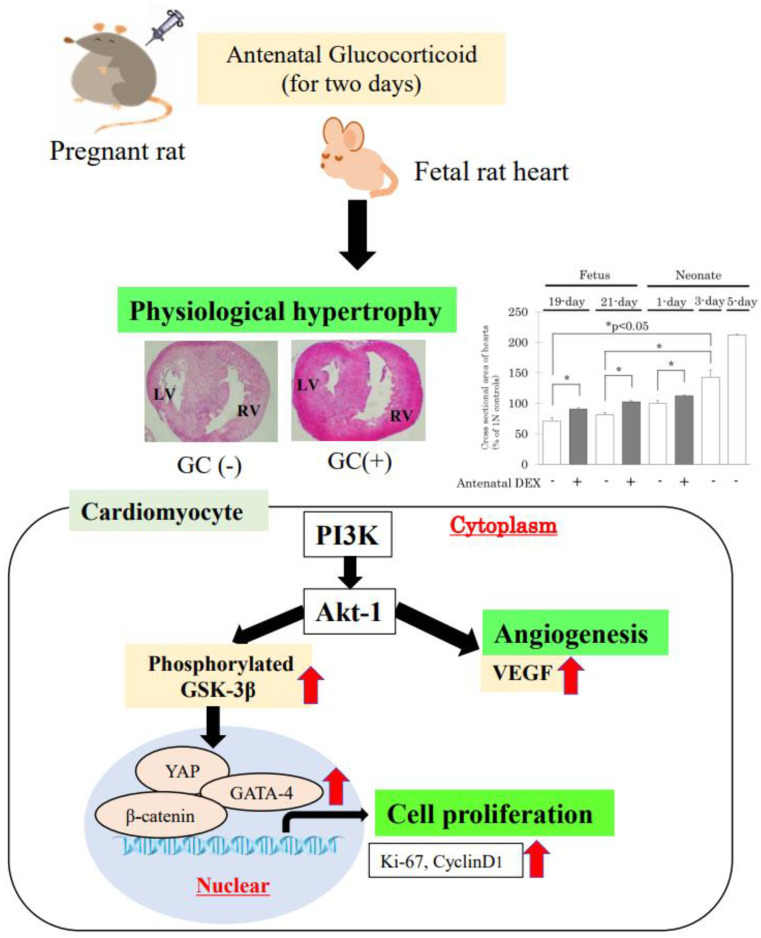
Schematic diagram of cell proliferation and angiogenesis. Pregnant rats are administered glucocorticoids (GCs) subcutaneously for two days, on gestational days 17 and 19, and fetuses are delivered by cesarean section. On histological analysis, the cross-sectional area of the myocardium is smaller in 19-day-old fetuses than in neonates. Physiological hypertrophy is induced by cardiac cell proliferation and angiogenesis. Antenatal GC administration enhances phosphatidylinositol-3 kinase (PI3K), which activates Akt-1. Moreover, Akt-1 accelerates both phosphorylated glycogen synthase kinase-3β (p-GSK-3β) and vascular endothelial growth factor (VEGF) expression. Subsequently, p-GSK-3β contributes to the activation of cardiomyocyte proliferation factors, β-catenin, Yes-associated protein (Yap), and GATA-4. Accordingly, cell proliferation markers, Ki-67 and cyclin D1, are increased in the nuclei of fetal cardiomyocytes. GATA binding protein, GATA.

**Figure 2 ijms-23-10186-f002:**
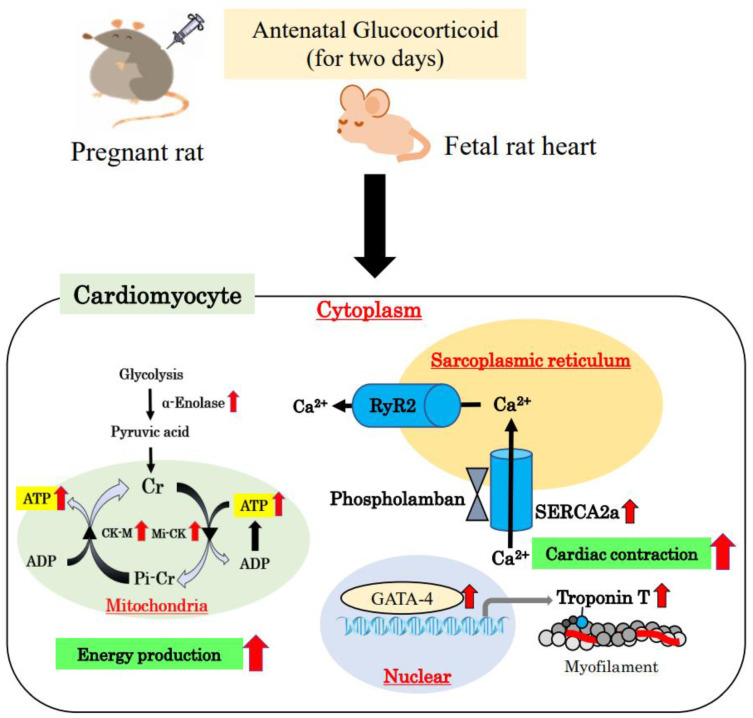
Schematic diagram of energy production and acceleration of cardiac contraction. Energy metabolism in the fetal heart is relatively dependent on anaerobic glycolysis. α-Enolase converts glucose into pyruvate. Energy released during glycolysis is used to make ATP, and antenatal glucocorticoid (GC) administration accelerates these pathways. Furthermore, antenatal GC administration induces creatine kinase (CK) protein production by promoting myofibrillar-bound M isoenzyme CK (CK-M) and mitochondrial CK (Mi-CK) gene expression in the mitochondria. Increased CK-M and Mi-CK enzymes contribute to increased ATP synthesis. Cardiac energy metabolism is essential for normal cardiac contractile function. GATA-4 has not only involvement in cell proliferation but also in binding the cells to the troponin T promoter. Antenatal GC increases the expression of GATA-4, troponin T, SERCA2a, and phospholamban. Cardiac contraction is increased with the increase in intracellular Ca^2+^ concentration and troponin T expression. SERCA2a plays a crucial role in the regulation of the intracellular Ca^2+^ concentration in the SR. Ca^2+^ enters myocytes during an action potential through voltage-gated Ca^2+^ channels in the sarcolemma. This Ca^2+^ influx triggers the release of Ca^2+^ from the SR by RyR2, which increases the intracellular Ca^2+^ concentration, resulting in myocardial contraction. Sarco-endoplasmic reticulum Ca-ATPase, SERCA; sarcoendoplasmic reticulum, SR; ryanodine receptor, RyR.

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
