# Peer review of "Antenatal Glucocorticoid Administration Promotes Cardiac Structure and Energy Metabolism Maturation in Preterm Fetuses"

_ijms, 2022, doi:10.3390/ijms231710186_

Round 1
Reviewer 1 Report
The manuscript by Sakurai et al. reviews potential effects of antenatal glucocorticoid administration in heart development. It includes an account of molecular mechanisms likely affecting glucocorticoid effects on heart development, cardiomyocyte energy metabolism and excitation/contraction coupling. It finishes with a view on current clinical aspects of antenatal glucocorticoid administration. While the subject is very interesting and has significant clinical impact, the review can be improved for readability and usability by researchers in the field. In addition, there are concerns regarding the usage of references in the text (see below).
General comments:
1. A large part of the discussion about clinical repercussions of antenatal glucocorticoid administration relies on reference 68 (Crowley, P. 2000). This is concerning, since this review was withdrawn in 2008 (apparently, to substitute it with a newer version). This strongly suggests that the account of the authors is outdated.
2. In many sections, the order in which facts are presented is confusing and does not help overall readability. For instance, the Introduction seems to go back and forth between the overall problem (risks associated to pre-term delivery and usefulness and side effects of glucocorticoid therapy) and issues directly related to heart development. It would be more useful to go from the general to the specific. The same happens in section 4, where the text quicly goes into details of SERCA2 function and only later explains the basics about heart excitation/contraction coupling and Ca2+ handling. Again, it would be more useful to start with the general facts (physiology of the heart excitation/contraction coupling), then general molecular mechanisms and then the specifics of the fetal/neonatal heart.
3. Important paragraphs are missing any references to published work. For instance, lines 36-38 and lines 297-304.
4. The review is imbalanced in the level of detail provided to the reader. Some important issues are glossed over without many specifics, while primary work where the authors have been involved described with extraordinary level of detail. While it is understandable that the authors wish to disseminate their own results, the review would benefit from a more balanced view of the field.
5. Important facts about glucocorticoid metabolism and signaling are overlooked in the review. The manuscript does not mention the different roles of receptors (glucocorticoid and mineralocorticoid receptor, GR and MR) in the heart or the possible interplay between them (MR is a high-affinity glucocorticoid receptor). Enzymes that potentially modify local glucocorticoid levels, either by synthesis or metabolism, are also not mentioned. In addition, the review focuses on downstream pathways (PI3K-Akt) while failing to mention significant variants of those pathways (SGK1 signaling instead of Akt).
Specific points:
· Line 30, eliminate “were”
· Line 36, eliminate “were”
· Line 44, substitute “Since” for “Since then”
· Lines 47-48, “…heart development has a highly dynamic process…” should be “…heart development is a highly dynamic process…”
· Line 89, the phrase seems to be wrong: “..in accommodate...”? In general, this paragraph needs re-writing, it is not easy to read in its present form.
· Line 97: “…we have investigated that dexamethasone…”, the authors probably mean that “…we have established that dexamethasone…”
· Line 164, the phrase affirms that enolase is “a high-energy intermediate in the process of energy production”, please explain the meaning of this sentence.
· Lines 166-167 contain an apparently unfinished sentence.
· Line 192, what do the authors mean by “progress of ATP synthesis”?
· In section 4, ensure that the abbreviation SR (sarcoplasmic reticulum) is consistently used after first definying it.
· Line 229, “intercellular”, the authors probably mean “intracellular”.
· Conclusions: the last phrase appears disconnected to the focus of the review. Conclusions should try to wrap up the main ideas of the paper, not introduce vague statements about the usefulness of the work in the field.
Author Response
A point by point replay to reviewer’s comments
In response to the comments by reviewer # 1
Thank you for your careful review of our manuscript. We appreciate to your helpful comments and advice for our paper. We have modified the manuscript according to reviewer’s advice as below. Furthermore, we corrected the format according to the new submission rules. The red characters are places of correction.

Reviewer 2 Report
The review paper is comprehensive and interesting for a broad number of readers.
There are some points that I suggest to consider:
1) the authors mention hypertrophy and hyperplasia of the cardiomyocyte component. In mammals, depending on the species, the cardiomyocyte undergo a limited number of mytosis after birth (during the first week), not necessariy followed by cytodieresis, most cells resulting polynucleated. How do GCs impact the nuclei number?
2) mammal cardiomyocytes are immature from several points of view, and correctly the authors mention cell metabolism and calcium homeostasis. They do not mention ECG, however QT prolongation and instability is a marker of the immature human heart during the first 2-3 weeks. Is there evidence of the impact of GCs supplementation?
3) When the authors mention the favorable impact of GCs on PDA, it is worth to recall that this effect might be related to decreased prostaglandins or other proinflammatory factors as cytokines. This leads to a general consideration: is the effect of GCs related (also) to its local anti-inflammatory effect?
Author Response
A point by point replay to reviewer’s comments
In response to the comments by reviewer # 2
Thank you for your careful review of our revised manuscript. We reread our revised manuscript and modified it in accordance with the reviewer’s comments. We have improved the stylistic expression of the revised paper and the manuscript had checked the grammar by a native English speaker. Furthermore, we have shown the blue characters are places of correction.
There are some points that I suggest to consider:
Q1) the authors mention hypertrophy and hyperplasia of the cardiomyocyte component. In mammals, depending on the species, the cardiomyocyte undergo a limited number of mytosis after birth (during the first week), not necessariy followed by cytodieresis, most cells resulting polynucleated. How do GCs impact the nuclei number?
Ans. Thank you for your comment. In common, the cardiomyocytes of fetal mammalian heart become postmitotic entering and lead to binucleation after birth. However, we considered the cardiomyocyte proliferative capacity may play a dominant role in the premature fetus with antenatal GC administration.
“In common, the cardiomyocytes of fetal mammalian heart become postmitotic enter-ing and lead to binucleation after birth. Whereas cortisol infused into the heart of fetal sheep had no effect on binucleation of cardiomyocytes, it induces cell proliferation. The cardiomyocyte proliferative capacity may play a dominant role in the premature fetus with antenatal GC administration.”
Giraud GD, Louey S, Jonker S, Schultz J, Thornburg KL. Cortisol stimulates cell cycle activity in the cardiomyocyte of the sheep fetus. Endocrinology. 2006 Aug;147(8):3643-9. doi: 10.1210/en.2006-0061. Epub 2006 May 11. PMID: 16690807.
Q2) mammal cardiomyocytes are immature from several points of view, and correctly the authors mention cell metabolism and calcium homeostasis. They do not mention ECG, however QT prolongation and instability is a marker of the immature human heart during the first 2-3 weeks. Is there evidence of the impact of GCs supplementation?
Ans. Thank you for your comments. There are not studies in the fetus, but there have been several reports showing the cases of prolonged QT interval with adrenal insufficiency. Previous reports described cases of Torsade de Pointes associated with hypopituitarism which were treated with steroid hormone. Considering that premature infants have reduced adrenal function, it is speculative that antenatal GCs administration may affect the QT interval.
Kim HN, Cho GJ, Ahn KY, Lee US, Kim HK, Cho HJ, Kim GH, Kim W, Jeong HM, Park CJ, Kang CJ (2001) A case of Torsade de Pointes associated with hypopituitarism due to hemorrhagic fever with renal syndrome. J Korean Med Sci 16:355–358
Kanamori K, Yamashita R, Tsutsui K, Hara M, Murakawa Y (2014) Long QT syndrome associated with adrenal insufficiency in a patient with isolated adrenocorticotropic hormone defieciency. Intern Med 53:2329–2331
Kang GD, Kim ES, Park SM, Kim JE, Lee HJ, Park GD, Han RK, Oh JD (2013) Acquired long QT syndrome manifesting with Torsades de Pointes in a patient with panhypopituitarism due to radiotherapy. Korean Circ J 43:340–342
Q3) When the authors mention the favorable impact of GCs on PDA, it is worth to recall that this effect might be related to decreased prostaglandins or other proinflammatory factors as cytokines. This leads to a general consideration: is the effect of GCs related (also) to its local anti-inflammatory effect?
Ans. Thank you for your comment. We searched studies in fetal heart relating to antenatal GC and inflammation, but there were few studies.
Fadenet et al have reported that antenatal glucocorticoids have not a sustained anti-inflammatory effect in infants born before 28 weeks of gestational age. However, this study has shown on effect of GC for prevention to systematic infection. Therefore, we didn’t mention about that in our review. We would like to consider the subject from now on.
*Faden M, Holm M, Allred E, Fichorova R, Dammann O, Leviton A; ELGAN Study Investigators. Antenatal glucocorticoids and neonatal inflammation-associated proteins. Cytokine. 2016 Dec;88:199-208. doi: 10.1016/j.cyto.2016.09.015.

Reviewer 3 Report
In this review titled “Antenatal glucocorticoid administration promotes cardiac structure and energy metabolism maturation in preterm fetuses” Sakurai and co-authors discussed the potential biochemical and physiological mechanisms of one of the most important antenatal therapies available to improve newborn outcomes in case of expected preterm birth – corticosteroid administration. In the introductory section, the authors provided a detailed description of the glucocorticoids use and the problems associated with it, especially in relation to the fetal heart development. In the main section, based on studies of laboratory animals, the authors proposed several biochemical models of the action of glucocorticoids on the metabolism, growth and development of cardiomyocytes in the fetal heart. In the concluding remarks of the review, the authors discussed the limitations of using animal models to uncover the mechanism of antenatal glucocorticoid therapy and future prospects in this area.
A overview of the literature related to the topic of this review shows that this is a very active field of science, and the effect of antenatal glucocorticoid therapy on the fetal heart remains unclear.
The manuscript is acceptably well written and has valuable and attractive ideas, but I have some major concerns that prevent me to endorse its acceptance at the present stage.
I would like to address several major issues concerning clearity and quality of presented manuscript.
Firstly, I should note that both figures in the article are of poor quality and have insufficient description in the figure legend. Both figures are labeled "figure" as part of the diagram. The figure elements are color-coded, figures have arrows and inserts of other figures without proper explanation, making it difficult to understand and interpret the meaning behind them. I would recommend completely redoing the figures.
Secondly, the authors seem to have failed to draw sufficient comparisons and parallels between data obtained from laboratory animal studies, clinical practice, and clinical guidelines. In section 5, only the second paragraph is devoted to describing the use of glucocorticoid therapy in clinical practice and does not include the comparison between treatment strategies for human fetuses and animal models.
Lastly, the second paragraph of section 5 mentioned above (lines 277-295) contains many direct citations from www.acog.com without a proper reference. I would like to suggest that this section be improved by presenting a comparative table for the current recommendations from different medical societies or medical regulators in different countries. It is also desirable to include in such a table data on outcomes and complications associated with cardiac function.
Minor issues:
Line 30: ‘were obtained’ -> ‘obtained’.
Lines 56-57: Sentence starting with ‘Endogenous’. Please clarify where you are writing about fetal and about maternal glucocorticoid concentrations.
Line 89: ‘in accommodate’.
Line 107: ‘(PI3K)-Atk pathway’ -> ‘PI3K-Atk pathway’.
Line 117: ‘While,’.
Lines 164-167: Incomprehensible sentence structure starting with ‘Enolase’.
Lines 299-301: Sentence starting with ‘Although’. Consider clarifying the this sentence. Perhaps, you meant ‘Although these models cannot reflect all clinical features,’.
Author Response
A point by point replay to reviewer’s comments
In response to the comments by reviewer # 2
Thank you for your careful review of our manuscript. We appreciate to your helpful comments and advice for our paper. We have modified the manuscript according to reviewer’s advice as below. Furthermore, we corrected the format according to the new submission rules. The red characters are places of correction.

Reviewer 4 Report
The review aim was to describe the fetal heart growth with antenatal GC administration in experimental animal models. The study shold be improved.
I suggest review design figure for the deep understanding the review objective. Beside this figure, I would like to know what is the novelty of this review in comparison with previous work.
In the reference part, the citation type is not satisfactory, doi numbers are missing, the authors should be improved it.
Author Response
A point by point replay to reviewer’s comments
In response to the comments by reviewer # 3
Thank you for your careful review of our manuscript. We appreciate to your helpful comments and advice for our paper. We have modified the manuscript according to reviewer’s advice as below. Furthermore, we corrected the format according to the new submission rules. The red characters are places of correction.

Round 2
Reviewer 1 Report
While the authors have answered most of my concerns, the corrections introduced in the text seem unconnected with the previous version. For instance, see lines 136-145, where an older phrase stands between two new sentences with no apparent connection. Also, the text still contains incoherent phrases (see for instance lines 133-135 or lines 322-323). Overall, I think the authors have not made enough effort to follow the reviewer´s comments to make the text more readable and improve its english usage. The review can still be signifcantly improved by another round of review, focusing on readibilty and the logical flow of the text.
Author Response
A point by point replay to reviewer’s comments
In response to the comments by reviewer # 1
Thank you for your careful review of our revised manuscript. As you point out, we apologize for our English expressions are clumsy. We reread our revised manuscript and modified it in accordance with the reviewer’s comments. We have improved the stylistic expression of the revised paper and the manuscript had checked the grammar by a native English speaker. Furthermore, we have shown the blue characters are places of correction.
- While the authors have answered most of my concerns, the corrections introduced in the text seem unconnected with the previous version. For instance, see lines 136-145, where an older phrase stands between two new sentences with no apparent connection. Also, the text still contains incoherent phrases (see for instance lines 133-135 or lines 322-323). Overall, I think the authors have not made enough effort to follow the reviewer´s comments to make the text more readable and improve its English usage. The review can still be significantly improved by another round of review, focusing on readability and the logical flow of the text.
Ans. We agree with your comment. We have reread the revised text and corrected the incoherent phrases in the section 2 and 5 of the 2nd revision.

Reviewer 3 Report
I thank the authors for their careful comments and revision of the manuscript. The concept of antenatal action of glucocorticoids on the fetal heart is novel and has scientific merit, although it might need further development and careful experimental verification.
During a detailed acquaintance with the corrected version of the manuscript, I did not see significant improvements in the quality of the material, and in some aspects the changes made seem to have been made hastily and did not bring clarity. Often the changes are not consistent with the surrounding text and stand out, as can be seen in sections 3 and 5, which impairs readability. Furthermore, the authors failed to properly address the issue of plagiarism form the www.acog.com in the second paragraph of section 5. The text is still 64% plagiarized, which is unacceptable and needs to be corrected regardless of added references. Additionally, the figures still contain the text ‘Figure 1’, ‘Figure 2’ as part of the scheme that are not necessary.
In general, the clarity of the manuscript is still hindered by minor, but scattered throughout the text, language errors. Unfortunately, there are still quite a lot of errors, and therefore it is difficult to list and analyze them all as a part of this review.
The list of references contains many articles (37 out of 79) that were published more than ten years ago, which also casts doubt on the relevance and validity of the issues discussed in the review. Section 3 is particularly affected as most of the references in it are over 20 years old.
Thus, the emerging contradictions between the reviewed material and the novelty of the proposed concepts do not allow me to consider this article as a review. At its core, this manuscript corresponds to the profile of an article in the field of theoretical biology and therefore, it seems to me, should be peer-reviewed in accordance with the criteria for a research article.
These factors do not allow me to conclude that specialized and narrow subject matter of this manuscript and its quality in the current form are consistent with the scope—fundamental theoretical problems of broad interest in biology, chemistry and medicine—of a highly ranked International Journal of Molecular Sciences and the expectations of its readers.
In this regard, I would like to recommend resubmitting this manuscript, after careful editing, to a more specialized MDPI journal such as Endocrines, or to this journal, as a grammatically correct and concise research article.
Author Response
A point by point replay to reviewer’s comments
In response to the comments by reviewer # 3
Thank you for your careful review of our revised manuscript. As you point out, we apologize for our English expressions are clumsy. We reread our revised manuscript and modified it in accordance with the reviewer’s comments. We have improved the stylistic expression of the revised paper and the manuscript had checked the grammar by a native English speaker. Furthermore, we have shown the blue characters are places of correction.
- During a detailed acquaintance with the corrected version of the manuscript, I did not see significant improvements in the quality of the material, and in some aspects the changes made seem to have been made hastily and did not bring clarity.
Often the changes are not consistent with the surrounding text and stand out, as can be seen in sections 3 and 5, which impairs readability.
Ans. We appreciate the reviewer's comment on this point. We reread our revised manuscript and modified it in accordance with the reviewer’s comments.
- Furthermore, the authors failed to properly address the issue of plagiarism form the www.acog.com in the second paragraph of section 5. The text is still 64% plagiarized, which is unacceptable and needs to be corrected regardless of added references.
Ans. We agree with your comment, and the text has corrected the second paragraph of section 5 in the 2nd revised manuscript.
- Additionally, the figures still contain the text ‘Figure 1’, ‘Figure 2’ as part of the scheme that are not necessary.
We apologize for our mistake. The unnecessary figures have deleted.
- In general, the clarity of the manuscript is still hindered by minor, but scattered throughout the text, language errors. Unfortunately, there are still quite a lot of errors, and therefore it is difficult to list and analyze them all as a part of this review.
Ans. As you point out, we apologize for our English expressions are clumsy. We reread our revised manuscript and modified it in accordance with the reviewer’s comments. We have improved the stylistic expression of the revised paper and the manuscript had checked the grammar by a native English speaker.
- The list of references contains many articles (37 out of 79) that were published more than ten years ago, which also casts doubt on the relevance and validity of the issues discussed in the review. Section 3 is particularly affected as most of the references in it are over 20 years old.
Thus, the emerging contradictions between the reviewed material and the novelty of the proposed concepts do not allow me to consider this article as a review. At its core, this manuscript corresponds to the profile of an article in the field of theoretical biology and therefore, it seems to me, should be peer-reviewed in accordance with the criteria for a research article.
Ans. We appreciate the reviewer's comment on this point. As your point out, we searched the published data relating to effect of energy metabolism by antenatal GC in section 3, but we couldn’t find new literatures except for our data. Therefore, we necessarily quoted the references published more than ten years ago. However, we think that novelty of the concepts from our studies has discussed. The text has corrected the section 3 in the 2nd revised manuscript.
- These factors do not allow me to conclude that specialized and narrow subject matter of this manuscript and its quality in the current form are consistent with the scope—fundamental theoretical problems of broad interest in biology, chemistry and medicine—of a highly ranked International Journal of Molecular Sciences and the expectations of its readers.
In this regard, I would like to recommend resubmitting this manuscript, after careful editing, to a more specialized MDPI journal such as Endocrines, or to this journal, as a grammatically correct and concise research article.
Ans. Thanks for your advice. We resubmit this revised manuscript after careful editing and grammatically correct. We will keep it in mind and try my best. We will entrust reviewer's concerns to International Journal of Molecular Sciences.

Reviewer 4 Report
The authors made modifications what I suggested.
Author Response
A point by point replay to reviewer’s comments
In response to the comments by reviewer # 4
Thank you for your valuable advices.
The final version after proofreading is attached.
